# Learning From Correctness Without Prompting Makes LLM Efficient Reasoner

**Yuxuan Yao**[*], **Biyan Zhou, Sichun Luo, Linqi Song**[†]
Department of Computer Science
City University of Hong Kong
Tat Chee Avenue, Kowloon, Hong Kong SAR
`yuxuanyao3-c@my.cityu.edu.hk, linqi.song@cityu.edu.hk`

**Han Wu**[*], **Zhijiang Guo**[†], **Jiahui Gao, Xiaojin Fu,**
Huawei Noah's Ark Lab
Hong Kong SAR
`{wu.han1, guozhijiang}@huawei.com`

**Hanxu Hou,**
Department of Communication Engineering
Dongguan University of Technology
`hhx@dgut.edu.cn`

## Abstract

Large language models (LLMs) have demonstrated outstanding performance across various tasks, yet they still exhibit limitations such as hallucination, unfaithful reasoning, and toxic content. One potential approach to mitigate these issues is learning from human or external feedback (e.g. tools). In this paper, we introduce an intrinsic self-correct reasoning framework for LLMs that eliminates the need for human feedback, external tools, and handcraft prompts. The proposed framework, based on a multi-step reasoning paradigm **Le**arning from **Co**rrectness (LECO), improves reasoning performance without needing to learn from errors. This paradigm prioritizes learning from correct reasoning steps, and a unique method to measure confidence for each reasoning step based on generation logits. Experimental results across various multi-step reasoning tasks demonstrate the effectiveness of the framework in improving reasoning performance with reduced token consumption. The code is available at `https://github.com/starrYYxuan/LeCo`.

## 1 Introduction

Large language models (LLMs; Brown et al. 2020; OpenAI 2023; Touvron et al. 2023) have exhibited remarkable performance on a diverse range of natural language processing benchmarks (Hendrycks et al., 2021a; Srivastava et al., 2022) and also showcased promising results on real-world applications (Wu et al., 2023; Thirunavukarasu et al., 2023). However, it is imperative to acknowledge that LLMs still possess certain limitations. For instance, the occurrence of undesirable behaviors like hallucinations (Rawte et al., 2023), generating harmful content (Bai et al., 2022), and non-adherence to established rules and constraints (Ouyang et al., 2022; Peng et al., 2023) remains largely unexplored.

One extensively employed approach to address these problems is learning from feedback (Pan et al., 2023). It involves guiding LLMs to improve their responses through a cycle of trial, examination, and correction. During the examination phase, feedback is provided to

---

[*]Equal Contribution.
[†]Corresponding Authors.

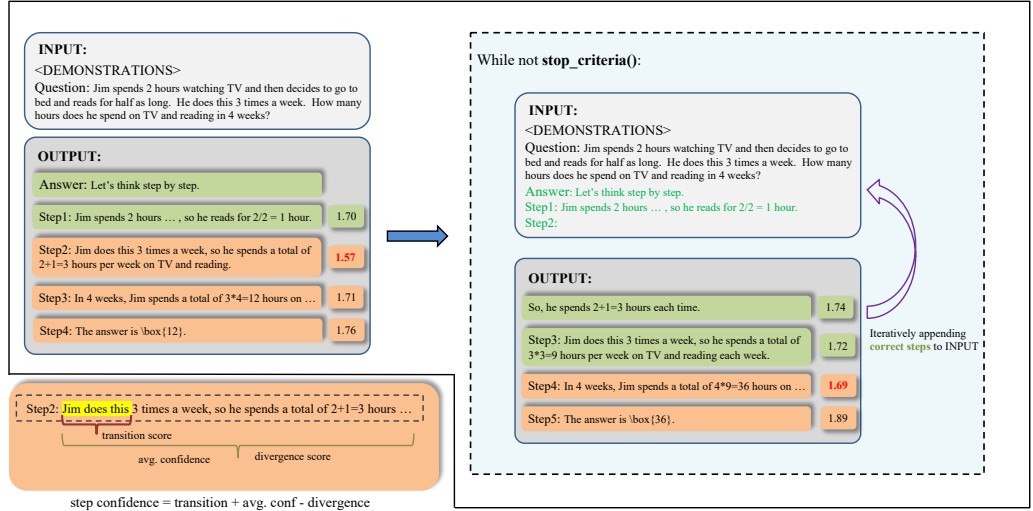

Figure 1: The framework of LECO. LECO first obtains an initial solution for the input problem. Then, we progressively collect the correct steps from the latest solution until the final answer is obtained.

identify the shortcomings in the trial answer and guide the necessary corrections. Prior efforts (Huang et al., 2023a; Gou et al., 2023a) have confirmed high-quality feedback can offer valuable insights into further corrections. Although human feedback (Ouyang et al., 2022; Fernandes et al., 2023) and external tools feedback (Gou et al., 2023a;b) are generally valuable, they are either expensive to collect or heavily dependent on the abilities of the selected tools. To eliminate external intervention, another popular line of research is self-correction, where the model progressively learns from the feedback it generates internally, without relying on external sources (An et al., 2023). However, Huang et al. (2023b) recently suggests that LLMs do not possess the inherent capabilities to find the errors and rectify their responses just by designing the prompts. More frustratingly, these methods often require creating extensive and elaborate handcraft prompts to guide the model in acquiring and understanding the feedback, which is a time-consuming and labor-intensive process, finally tuning our researchers into "prompt engineers".

In this work, we present a novel intrinsic self-correct reasoning framework that eliminates the need for human feedback, external tools, and handcraft prompts. Different from the existing self-correction methods, which are predominantly based on learning from errors (An et al., 2023; Gou et al., 2023a), we propose a new multi-step reasoning paradigm known as **Le**arning from **Co**rrectness (LECO). As illustrated in Figure 1, we begin by assigning a confidence score to each reasoning step in the first-round reasoning path. The step with the lowest confidence score will be identified as the earliest potential error step, and the steps before this point are considered to be "correct". Then, the correct steps, considered as "correctness", are appended to the input, and repeat the reasoning process. While the insight of learning from errors comes from the learning process of human students, the motivation behind our method is derived from progressive learning (Wu et al., 2019; Fayek et al., 2020), where correct reasoning steps are gradually accumulated to ultimately approach the correct answer. Furthermore, we also introduce an efficient method to measure the confidence for each reasoning step based on the generation logits, without the need for additional tokens or external tools. Specifically, we jointly consider the average confidence of each token within a step, the confidence divergence of a step, and the probability of step transmission to calculate the overall step confidence. We surprisingly find our method can identify almost 65% incorrect steps. We conduct experiments with both closed-source models (e.g. GPT-3.5 and GPT-4) and open-source models (e.g. DeepSeek; Shao et al. 2024) on various multi-step reasoning tasks, including arithmetic reasoning, commonsense reasoning, and logical reasoning, show that our framework can significantly improve reasoning performance with less token consumption.

Our primary contributions include 1) we propose a novel multi-step reasoning paradigm learning from correctness, dubbed as LeCo, which progressively accumulates the correct steps and approaches the final answer; 2) we challenge the conventional belief that high-quality feedback can only come from external sources and propose a unique intrinsic method to measure the confidence for each reasoning step, and 3) Both the off-the-shelf and open-source models can benefit from LeCo on various multi-step reasoning tasks with reduced token consumption. More excitingly, LeCo completely eliminates the need for prompt engineering.

## 2   Related Work

**Learning from Feedback**   Improving LLMs through learning from feedback has become a prevalent strategy, notably through reinforcement learning from human feedback, which seeks to align LLMs with human values by refining their outputs based on feedback (Ouyang et al., 2022; Bai et al., 2022; Touvron et al., 2023). However, this method faces challenges such as high costs due to manual labor and a lack of real-time feedback capabilities (Pan et al., 2023; Fernandes et al., 2023). An alternative strategy involves using self-correcting LLMs, which rely on automated feedback to iteratively adapt and understand the consequences of their actions without heavy reliance on human intervention. This feedback can be derived from outside sources such as other models (Yang et al., 2022; Lightman et al., 2023; Xiong et al., 2023), tools (Huang et al., 2024; Lu et al., 2024b), knowledge bases (Gao et al., 2023; Yu et al., 2023), or evaluation metrics (Jung et al., 2022; Welleck et al., 2023).

External feedback leverages external perspectives to identify errors and verify factual accuracy, offering insights that may not be recognized by the LLM alone. Conversely, feedback can also be internally generated, where the LLM evaluates and refines its output iteratively until a desired quality is achieved (Madaan et al., 2023; Shinn et al., 2023; Helbling et al., 2023; Xie et al., 2023). This self-improvement mechanism is particularly valuable in scenarios where external feedback is scarce or restricted (Yan et al., 2023; Lu et al., 2024a). However, Huang et al. (2023b) suggests that LLMs struggle to independently identify and correct errors through self-generated prompts. Recent effort (Gonen et al., 2023) show that an LLM's familiarity with a prompt's language predicts its effectiveness, with lower perplexity prompts leading to better performance. Unlike existing efforts, LeCo focuses on learning from one's correct reasoning steps, without the need for feedback mechanisms including human intervention, external tools, or tailored prompts.

**Reasoning without Prompting**   Recent studies have been focusing on improving the reasoning abilities of LLMs through various methodologies, primarily centered around the enhancement of prompting techniques. These works include few-shot prompting with intermediate steps augmented demonstrations (Wei et al., 2022; Fu et al., 2023; Yao et al., 2023; Wang et al., 2023) or zero-shot prompting with specific instructions (Kojima et al., 2022; Yasunaga et al., 2023). Although these methods have shown promising results, their effectiveness is often constrained by their task-specific nature and the labor-intensive process of designing prompts, leading to inconsistent outcomes across different tasks (Ye & Durrett, 2022; Zhou et al., 2023).

Another strategy to facilitate reasoning involves instruction tuning, which leverages a significant volume of chain-of-thought (CoT) data (Chung et al., 2022; Mukherjee et al., 2023; Gunasekar et al., 2023; Luo et al., 2023). Recently, Liu et al. (2024) proposed to tune LLMs by comparing the logit differences between a pair of tuned and untuned smaller models, showcasing improvements in reasoning without CoT distillation. In contrast to these methods, our LeCo introduces an intrinsic self-correct reasoning mechanism that does not depend on fine-tuning or auxiliary models.

Additionally, there has been an interest in refining decoding algorithms specifically for reasoning. Notably, contrastive decoding (Li et al., 2023) has been developed to enhance a model's generation quality by adjusting the logits from smaller models, with recent research indicating its potential to boost reasoning performance (O'Brien & Lewis, 2023). Wang & Zhou (2024) discovered that CoT reasoning patterns naturally occur within the

decoding trajectories of LLMs, leading to the development of CoT-decoding, which aims to identify more reliable decoding paths. Such advancements present a promising avenue to augment the efficacy of LECO. Future work could explore the integration of these decoding algorithms to extend beyond the current use of greedy decoding.

# 3 Methodology

We introduce LECO, a learning from correctness framework, designed to enhance multi-step reasoning capabilities. Our core insight is that providing the model with more correct reasoning steps helps it narrow down the search space for the solution. This facilitates the process of reaching the final answer. To achieve this, LECO utilizes a prompt-free method to calculate the confidence score of each reasoning step. By identifying the most reliable steps, the model can then leverage these insights to guide its reasoning process.

## 3.1 Step Confidence

**Preliminary** In generation tasks, logits represent the log probabilities of candidate tokens being chosen as the next word. Confidence, on the other hand, refers to a model's certainty in its prediction. Within reasoning tasks, step confidence specifically measures the model's belief in the correctness or factual basis of each reasoning step. Inspired by Li et al. (2023), we propose leveraging logits to estimate step confidence. We further design three logit-based scores that comprehensively evaluate confidence from both intra- and inter-step perspectives.

---

**Algorithm 1** Confidence-based Reasoning Algorithm

---

**Require:** input $x_0$, model $M$, demonstration $Demo_x$, stop condition $stop(*)$
1: $y_0 = \mathcal{M}(x_0, Demo_x)$           ▷ Initial Generation (Eq.5)
2: **for** iteration t $\in 1, \ldots, t$ **do**
3:     **if** not $stop(y_t)$ **then**           ▷ Stop Condition
4:        **for** step i $\in 0, \ldots, |y_0|$ **do**
5:           $s_e = Lowest(s_i\_score)$           ▷ Lowest Confidence Step (Eq.4)
6:        **end for**
7:        $x_t \leftarrow x_{t-1} + y_{t-1}(s < e)$
8:     **end if**
9:     $y_{t+1} = \mathcal{M}(x_t, Demo_x)$           ▷ Rethink Generation
10: **end for**
11: **return** $y_t$

---

Formally, we denote the entire reasoning path as $S = (s_1, s_2, \ldots, s_n)$, consisting of $n$ individual steps. Each reasoning step $s_i = \left(t_{i,1}, t_{i,2}, \ldots, t_{i,|s_i|}\right)$ is a sequence of tokens. We then apply the `Softmax` function on the logits score to obtain the probabilities $p_{i,j}$ for each token $t_{i,j}$.

**Average Token Score** A straightforward approach to measure step confidence is by averaging the token probabilities within a given step. This average reflects the model's certainty in its reasoning during that step. Therefore, we define single-step confidence as:

$$avg\_score_i = \frac{1}{|s_i|} \sum_{j=1}^{|s_i|} p_{i,j} \tag{1}$$

**Step Divergence Score** While average token probability seems intuitive, it can be misleading. Within a step, most tokens tend to be common words with high confidence scores but carry little information. Conversely, tokens crucial for reasoning, e.g. mathematical

calculations, often have lower confidence. This paradox leads to a high average token confidence for the entire step, which contradicts our goal.

To address this issue, we propose the step divergence score. This metric measures the distribution uniformity of token probabilities within a step. Ideally, we want the token probabilities to be both high and evenly distributed across all tokens. To achieve this, we formulate the step divergence score based on the Kullback-Leibler Divergence (KLD; Kullback & Leibler 1951) between the normalized distribution $P_i = \text{norm}(p_{i,1}, p_{i,2}, ..., p_{i,|s_i|})$ of the token probabilities and the uniform distribution $U$:

$$diver\_score_i = \ln(\text{KLD}^\tau(P_i, U) + 1), \qquad (2)$$

where $\tau$ is the rescaling temperature for the KL divergence value, as the step divergence score is expected to vary between 0 and 1. In this work, $\tau$ is set to 0.3.

**Inter-step Transition Score**   Following the intra-step measurements, we sought to quantify the transition between consecutive steps. Our preliminary experiments yielded two key insights: 1) steps with lower overall confidence tend to have lower confidence levels specifically in the initial heading tokens (typically the first three), more dicussions can be found at Section D. 2) These initial heading tokens were also the most likely to change across different program runs. Based on these observations, we propose using the probabilities of the heading tokens in a step to represent the inter-step transition score between that step and the subsequent one. In other words, the transition score is determined by:

$$trans\_score_i = \frac{1}{K} \sum_{j=1}^{K} p_{i,j} \qquad (3)$$

where $K$ is set to 3 here. Further analysis of hyperparameter settings is discussed in Section C.

Overall, the confidence score $s_i\_score$ of step $s_i$ is denoted as,

$$s_i\_score = avg\_score_i + trans\_score_i - diver\_score_i \qquad (4)$$

### 3.2   LECO: Learning From Correctness

While leveraging step confidence scores, previous approaches (Gou et al., 2023a; Huang et al., 2023a) heavily rely on prompting LLMs to pinpoint and rectify erroneous steps. This dependence on prompts makes them rather sensitive. Our LECO framework tackles this issue by iteratively gathering correct steps and consequently refining the search space for potential reasoning steps. As depicted in Figure 1, LECO operates in a two-stage process.

**Initial Stage**   Given an input $x_0$ and the corresponding demonstrations $Demo_x$, the model $M$ generates an initial answer $y_0$:

$$y_0 = \mathcal{M}(x_0, Demo_x), \qquad (5)$$

where $y_0(s_0, s_1, ..., s_{|y_0|})$ consists of multiple reasoning steps.

**Rethink Stage**   In this stage, we first calculate the confidence score for each step within the initial solution $y_0$ based on Eq. 4. We take the step with the lowest step confidence or the earlier one of the two steps with the lowest step confidence as the earliest error step, which depends on the complexity of the reasoning problems. Denote the selected error step as $s_e, 1 \leq e \leq |y_0|$[1], we name the steps before $s_e$ as "correctness" ($s_{<e}$). Then we iteratively append the correctness to the input and repeat the reasoning process with LLMs. At $t$-th iteration, the workflow can be formulated as,

---

[1]We always use "Let's think step by step." (Kojima et al., 2022) as the first step of the reasoning path and we do not consider the step confidence of this sentence.

| Model | Method | Date | Commonsense | | Arithmetic | | | Avg. |
|---|---|---|---|---|---|---|---|---|
| | | | CSQA | StrategyQA | AQuA | SVAMP | GSM8K | |
| GPT-3.5 | CoT | 80.80 | 79.69 | 73.25 | 51.57 | 84.00 | 77.86 | 74.53 |
| | Complex | 84.20 | 77.33 | 69.84 | 54.49 | 81.25 | 80.89 | 74.67 |
| | ADPSC | 83.60 | 75.92 | 68.99 | 51.97 | 78.89 | 79.00 | 73.06 |
| | SC | 84.48 | 77.47 | 70.37 | 55.51 | 81.6 | 81.03 | 75.08 |
| | RCI | 74.97 | 68.34 | 51.94 | 35.50 | 79.95 | 75.25 | 64.33 |
| | LECO+CoT | 82.8 | 79.77 | 71.13 | 52.72 | 85.00 | 78.24 | 74.93 |
| | | (+2.00) | (+0.08) | (-2.12) | (+1.15) | (+1.00) | (+0.38) | (+0.40) |
| | LECO+Complex | 84.92 | 77.68 | 71.05 | 56.77 | 82.35 | 82.33 | 75.85 |
| | | (+0.72) | (+0.35) | (+1.21) | (+2.28) | (+1.10) | (+1.44) | (+1.18) |
| GPT-4 | CoT | 92.80 | 87.46 | 83.63 | 71.60 | 93.05 | 94.84 | 87.23 |
| | Complex | 90.40 | 86.40 | 82.75 | 71.94 | 90.90 | 95.42 | 86.30 |
| | ADPSC | 89.20 | 85.67 | 83.87 | 70.08 | 88.99 | 94.09 | 85.32 |
| | SC | 90.72 | 86.81 | 83.75 | 72.19 | 93.49 | 95.51 | 86.67 |
| | RCI | 89.88 | 86.16 | 74.62 | 47.59 | 90.59 | 86.23 | 79.18 |
| | LECO+CoT | 93.60 | 87.63 | 83.25 | 71.99 | 93.55 | 95.14 | 87.53 |
| | | (+0.80) | (+0.17) | (-0.38) | (+0.39) | (+0.50) | (+0.30) | (+0.30) |
| | LECO+Complex | 90.80 | 86.90 | 83.97 | 72.33 | 91.40 | 95.68 | 86.85 |
| | | (+0.40) | (+0.50) | (+1.22) | (+0.39) | (+0.50) | (+0.26) | (+0.55) |

Table 1: Performance of GPT models on logical reasoning, commonsense reasoning, and arithmetic reasoning tasks.

| Model | Method | Subset | | | | | | | Avg. |
|---|---|---|---|---|---|---|---|---|---|
| | | Algebra | Count | Geometry | Iter | Num | Prealgebra | Precaculus | |
| GPT-3.5 | Complex | 58.55 | 30.80 | 29.83 | 17.46 | 31.93 | 61.11 | 15.39 | 35.01 |
| | ADPSC | 54.22 | 28.18 | 26.89 | 13.69 | 28.93 | 59.70 | 14.34 | 32.28 |
| | SC | 56.20 | 30.87 | 29.98 | 17.65 | 32.25 | 61.80 | 18.13 | 35.27 |
| | RCI | 49.79 | 24.25 | 18.76 | 10.16 | 25.09 | 53.71 | 13.08 | 27.83 |
| | LECO+Complex | 58.72 | 34.70 | 31.89 | 18.80 | 33.37 | 62.21 | 18.53 | 36.89 |
| | | (+0.17) | (+3.90) | (+2.06) | (+1.34) | (+1.44) | (+1.10) | (+3.14) | (+1.88) |
| GPT-4 | Complex | 69.06 | 50.32 | 38.62 | 25.33 | 46.39 | 76.98 | 28.23 | 47.85 |
| | ADPSC | 60.13 | 40.13 | 30.55 | 15.84 | 37.39 | 69.46 | 21.10 | 39.23 |
| | SC | 71.04 | 52.23 | 40.48 | 25.89 | 50.37 | 77.84 | 30.51 | 49.77 |
| | RCI | 65.49 | 46.93 | 29.71 | 16.56 | 43.68 | 73.99 | 27.07 | 43.35 |
| | LECO+Complex | 71.92 | 53.27 | 41.13 | 27.49 | 49.14 | 78.29 | 32.02 | 50.47 |
| | | (+2.86) | (+3.05) | (+2.51) | (+2.16) | (+2.75) | (+1.31) | (+3.79) | (+2.62) |

Table 2: Performance of GPT models on the MATH dataset.

$$x_t \leftarrow x_{t-1} + y_{t-1}(s < e), \quad y_t = \mathcal{M}\left(x_t, Demo_x\right). \tag{6}$$

LECO alternates between input updating and rethink response generation until the stopping condition is met. The process either stops at a maximum iteration number $T$ or identifies the two consecutive same answers. The algorithm can be found in Algorithm 1.

## 4 Experiments

**Dataset and Baselines** We evaluate the performance of LECO using a variety of datasets and baselines. The datasets are categorized into three reasoning types: arithmetic reasoning, commonsense reasoning, and logical reasoning. The arithmetic reasoning datasets include GSM8K (Cobbe et al., 2021), MATH (Hendrycks et al., 2021b), AQuA (Ling et al., 2017), and SVAMP (Patel et al., 2021). For commonsense reasoning, we use CSQA (Saha et al., 2018) and StrategyQA (Geva et al., 2021). The logical reasoning dataset is represented by Date Understanding (Srivastava et al., 2022).

Our evaluation utilizes both off-the-shelf models, such as GPT-3.5-Turbo and GPT-4, and open-source models like DeepSeekMath-RL-7B (Shao et al., 2024). The open-source models are chosen for their superior performance on well-known mathematical datasets. We also

| Model | Methods | GSM8K | MATH | | | | | | | | Avg. |
|---|---|---|---|---|---|---|---|---|---|---|---|
| | | | Algebra | Count | Geometry | Iter | Num | Prealgebra | Precaculus | |
| DeepSeek | Complex | 79.76 | 69.96 | 40.08 | 38.41 | 21,59 | 40.56 | 68.35 | 24.18 | 47.87 |
| | LeCo+Complex | 80.14 | 70.51 | 40.30 | 38.62 | 22.15 | 42.69 | 68.52 | 23.99 | 48.37 |
| | | (+0.38) | (+0.55) | (+0.22) | (+0.21) | (+0.56) | (+2.13) | (+0.17) | (-0.19) | (+0.50) |

Table 3: Performance of DeepSeekMath-7B on GSM8K and MATH, where Count represents counting and probability subset; Iter refers to intermediate algebra subset; Num means number theory subset.

incorporate two suites of public demonstrations, namely exemplars from vanilla CoT (Wei et al., 2022) and exemplars from complex-CoT (Complex; Fu et al. 2023), which are prompts with higher reasoning complexity to improve language models multi-step reasoning ability.

We compare LeCo with several baselines, including self-consistency (SC; Wang et al. 2023), adaptive self-consistency (ADPSC; Aggarwal et al. 2023), and RCI (Kim et al., 2023). SC polls the LLM multiple times and outputs the most frequent solution. ADPSC follows SC manner while conserving iterations via dynamically adjusting the number of samples per question using a lightweight stopping criterion. RCI is a representative work of learning from errors, which identifies errors and then self-corrects using designed prompts. In most runs, we use greedy decoding with a temperature of 0, except for the adaptive self-consistency and self-consistency settings, where a temperature of 0.7 is applied. The iteration number of self-consistency is set to 10. All experiments are run 10 times with different seeds, and the average scores are reported.

**Main Results** As shown in Table 1, 2 and 3, LeCo consistently improves the reasoning performance across the board. Particularly noteworthy is its outstanding performance in arithmetic reasoning, especially evident in the MATH dataset. The MATH dataset is renowned for its challenging nature, like more intricate problems and the need for more reasoning steps, with common CoT approaches demonstrating limited effectiveness on this benchmark. However, LeCo effectively addresses this complexity by progressively collecting correct steps, thereby reducing reasoning perplexity and achieving substantial improvements. We also find that high-quality demonstrations are preferred when using LeCo as larger improvements are consistently observed with LeCo+Complex.

For commonsense reasoning tasks, LeCo obtains slight improvements or comparable performance against baselines. Except for the StrategyQA dataset, some performance drops are spotted. We think this is because commonsense reasoning necessitates incorporating knowledge concerning events and their relationships. However, LeCo primarily focuses on augmenting intrinsic reasoning ability through correctness, hence a moderate enhancement is deemed reasonable. This finding is also aligned with observations in Lyu et al. (2023). Conversely, remarkable improvements are obtained in the date understanding dataset since this task is more similar to mathematical reasoning. It is worth noting that the difficulty of the task correlates positively with the impact of LeCo, as evidenced by the substantial improvements achieved on the AQuA and MATH datasets. The primary reason for this is that the LLM tends to remain their initial reasoning path on the easy problems, offering fewer improvement rooms for LeCo. For a comprehensive evaluation, we also apply LeCo on the open-source model. We chose DeepSeekMath-RL-7B, as it demonstrates competitive performance in mathematical reasoning tasks. As shown in Table 3, LeCo can consistently improve the reasoning performance on GSM8K and MATH datasets, indicating its effectiveness on open-source models.

On the other hand, LeCo also exhibits its superiority in reducing token consumption. As shown in Section A.2, although adaptive self-consistency has tried to reduce the iterations and token consumption by settings the early stop criterion, it still needs almost 4.46 rounds to determine the final answer while RCI needs 2.74 rounds. However, using the similar stop criterion of RCI, LeCo can reach the final answer just with 2.15 rounds. This phenomenon suggests that learning from correctness is more effective than learning from errors, as it does not necessitate the model's understanding of the error cues. Additionally, during each iteration, LeCo reduces API consumption by alleviating prompting the model to identify

| Models | Methods | Datasets | |
|---|---|---|---|
| | | GSM8K | StrategyQA |
| GPT-3.5 | Complex | 82.47 | 70.17 |
| | Random | 82.09 | 69.96 |
| | | (-0.38) | (-0.21) |
| GPT-4 | Complex | 95.34 | 82.69 |
| | Random | 95.22 | 83.37 |
| | | (-0.12) | (+0.68) |

Table 4: Coarse-grained level ablation study on GSM8K and StrategyQA datasets with GPT-3.5.

| GSM8K | Exact Correct | Partial Correct | Wrong |
|---|---|---|---|
| Only Avg | 38 | 9 | 53 |
| Only Div | 35 | 16 | 49 |
| Only Trans | 42 | 24 | 34 |
| Avg+Div | 36 | 14 | 50 |
| Avg+Trans | 50 | 16 | 34 |
| Div+Trans | 47 | 16 | 37 |
| LeCo | **53** | 10 | 37 |

Table 5: Fine-grained level ablation study of the three factors for calculating the step confidence. Avg denotes the average token confidence; Div denotes the step divergence score; and Trans denotes the inter-step transition score.

and understand the errors and shortening the output length. Therefore, as shown in Section A.1, LeCo reduces the token consumption by 80%/20% compared to SC/RCI.

# 5 Further Analyses

**Ablation Study** We conduct ablation studies at two levels of granularity. At the coarse-grained level, we explore the effectiveness of the learning-from-correctness framework by replacing the selection of correct steps with random choices. Specifically, in the rethink stage, we randomly choose a reasoning step as the earliest error step and consider the preceding steps as the "correctness". From Table 4, we can see that the random selection of correct steps generally hurt the reasoning performance, suggesting the importance of identifying the true correctness.

At the fine-grained level, we deeply investigate the design of step confidence, which involves calculating the sum of the average token confidence, step divergence score, and inter-step transition score. To minimize the time and token consumption, we employ the accuracy of identifying the earliest error step as our metric. This measurement has proven to be crucial for enhancing reasoning performance in subsequent rounds, as evidenced by the results in Table 4. To this end, we randomly sampled 100 incorrect solutions on the GSM8K dataset and manually annotated the earliest error step for these solutions. Then, we divide the predicted step into three categories, including *exact_correct*, *partial_correct* and *wrong*, wherein *exact_correct* means the predicted step is exactly the labeled earliest step; *partial_correct* means the predicted step is an error step but located after the earliest step, and *wrong* means the predicted step is before the target location. As presented in Table 5, LeCo performs best in finding the earliest error step, with accuracy over 50%. We also observe the significant performance drops when separately adopting one of these factors. More interestingly, among the three factors, we find the inter-step transition score affects the final performance most. This finding is also well-aligned with the observations in our preliminary experiments, as stated in Section 3.1, which suggests that the heading tokens of a step warrant more attention.

**Rethink Analysis** As LeCo and RCI are both the self-refinement framework, distinguished by their learning mechanisms from correctness or errors, we then compare them regarding the changes in answers after the rethinking stage. As illustrated in Figure 2, on the GSM8K dataset, over 85% of the time, both LeCo and RCI retain the original answer. Among the remaining instances, LeCo can modify more incorrect answers to correct ones than RCI (3.7% vs. 1.5%). On the StrategyQA dataset, the performance gap between LeCo and RCI is more significant, where RCI revises 24.8% correct answers to incorrect. This phenomenon is in line with the recent findings(Huang et al., 2023b) that LLMs are currently incapable of self-correction based on their own feedback. Superior to RCI, LeCo cleverly uses the accumulated correct information and avoids meticulous self-evaluation prompts to achieve better reasoning performance.

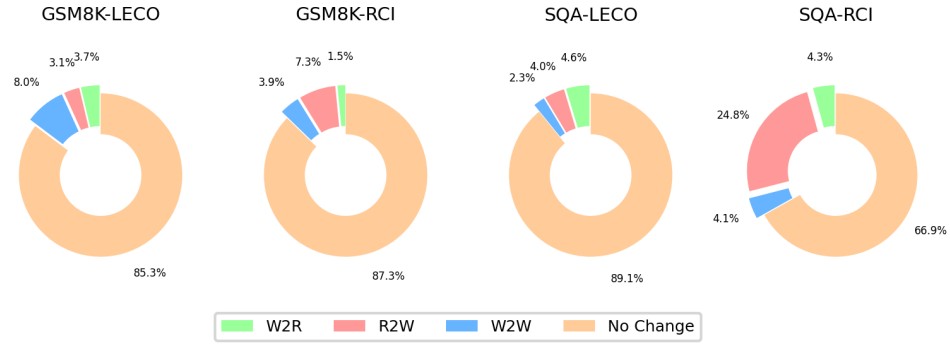

Figure 2: Evaluation of the changes after the rethink stage. We compare our LECO and RCI on GSM8K and StrategyQA datasets with GPT-3.5. W2R: the wrong answer is changed to right. R2W: the right answer is altered to wrong. W2W: a wrong answer is changed to another wrong answer. No change: The answer remains unchanged.

| Methods | Dataset | |
| --- | --- | --- |
| | StrategyQA | GSM8K |
| Complex | 31 | 10 |
| RANDOM | 25 | 13 |
| ORACLE | 36 | 22 |
| LECO | 33 | 21 |

Table 6: Oracle test on StrategyQA and GSM8K by GPT-3.5-Turbo. RANDOM denotes randomly selecting the earliest error step. ORACLE denotes human annotated earliest error step.

| Models | Methods | Datasets | |
| --- | --- | --- | --- |
| | | GSM8K | StrategyQA |
| GPT-3.5 | Complex | 81.58 | 70.94 |
| | Early stop | 82.03 (+0.45) | 69.31 (-1.63) |
| GPT-4 | Complex | 95.11 | 81.25 |
| | Early stop | 95.41 (+0.30) | 81.87 (+0.62) |

Table 7: Early Stop of LECO on the GSM8K and StrategyQA using GPT-3.5-Turbo and GPT-4.

**Oracle Test** We also conduct the oracle test to explore the upper bound of learning-from-correctness by directly providing the correct steps to LLMs during the rethink stage. To this end, we sampled 100 incorrect solutions generated by GPT-3.5-Turbo on the StrategyQA and GSM8K datasets, respectively. Subsequently, we manually annotate the earliest error step for these solutions. After collecting the preceding correct steps and appending them to the input, we generate an updated solution. As shown in Table 6, promising results are obtained that 36% and 22% wrong solutions can be amended with the help of correctness. It is important to note that these figures do not represent the absolute upper limit of the potential to learn from correctness since the refinement process is iterative but we can only label the first round. More interestingly, LECO achieves a comparable performance (33 vs. 36; 21 vs. 22) with ORACLE and significantly outperforms the random choices, suggesting the effectiveness of LECO in identifying the true correctness.

**Early Stop of LECO** As discussed above, the majority of initial solutions would not be modified after the rethink stage, which additionally escalates token consumption and ratio of "correct ⇒ incorrect". To alleviate these problems, we present an early stop strategy of LECO, which dynamically determines whether the initial solution requires refinement based on the overall solution score.

Similar to the step confidence, we calculate the overall solution confidence score $sln\_score$ by jointly considering the average score of step confidence and the inter-step divergence, formulated as,

$$sln\_score = \frac{1}{|sln|} \sum_{i=1}^{sln} s_i\_score - sln\_diver, \quad (7)$$

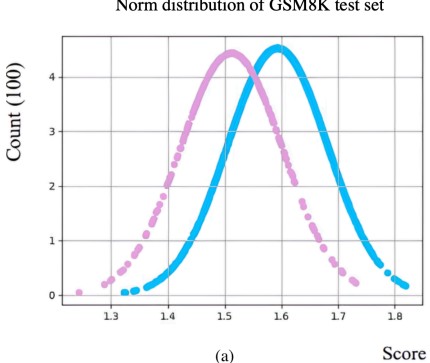 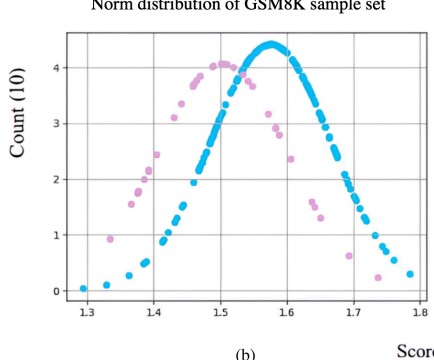

(a) Score          (b) Score

Figure 3: The distribution of correct and incorrect solutions of GSM8K by GPT-3.5-Turbo. The curve in pink represents incorrect answers, and the curve in blue represents correct answers.

where $s_i\_score$ is the confidence score of $i$-th step, obtained by Equation 4. $sln\_diver$ denotes the KL divergence between the normalized step scores $S = \mathrm{norm}(s_1\_score, ..., s_{|sln|}\_score)$ and an equal-length uniform discrete distribution, analogy to the Equation 2.

Firstly, we conducted the test on the GSM8K dataset using GPT-3.5-Turbo and recorded the solution confidence scores following Equation 7. As shown in Figure 3(a), we observed that the distributions of scores for both correct and incorrect solutions consistently tend to follow the norm distribution, with the average point of correct answers notably surpassing that of incorrect ones. We aim to employ this discrepancy to early stop the rethink stage. Specifically, we first randomly sample a subset from the testing data to obtain the distribution of solution scores, approximately 1/6 of the data of the entire test set used. Figure 3(b) illustrates the distribution on the GSM8K sample set, which also follows the norm distribution. Then, based on the 3-$\sigma$ characteristics of the norm distribution, we adopt the positive 1-$\sigma$ value from the score distribution of the incorrect solutions ($\mu + \sigma$) as our threshold, which covers 84% incorrect samples while only including around 50% correct instances.

As demonstrated in Table 7, consistent improvements can be obtained with early-stop LECO over the vanilla CoT-based method. Compared to the standard LECO, there are slight performance drops since more incorrect instances are filtered and not modified. However, early-stop LECO can still maintain the performance levels intermediate to those of SC and LECO while using fewer iteration rounds and tokens, approximately further reducing 10% tokens against the standard LECO (More details in Appendix B). We note that early-stop LECO is an alternative choice for the users to achieve a better trade-off between token consumption and performance.

## 6   Conclusion and Future Work

This work introduces LECO, an intrinsic self-correct reasoning framework designed to enhance LLM reasoning performance without relying on human feedback, external tools, or handcrafted prompts. LECO leverages a multi-step reasoning paradigm, prioritizing learning from successful reasoning steps. It incorporates a novel method for measuring confidence in each step based on generation logits. Our experiments across diverse multi-step reasoning tasks demonstrate LECO's effectiveness in improving reasoning accuracy while minimizing token consumption. This approach represents a distinct pathway for augmenting LLM capabilities, offering a promising avenue for advancing their aptitude in reasoning tasks. For future work, a worthy noting point is that LECO, especially its step confidence algorithm, would stand as an excellent candidate for pruning the complex reasoning structures, such as Tree-of-Thoughts (Yao et al., 2023) and Graph-of-Thoughts (Besta et al., 2023).

Acknowledgments

This work was supported in part by the Research Grants Council of the Hong Kong SAR under Grant GRF 11217823 and Collaborative Research Fund C1042-23GF, the National Natural Science Foundation of China under Grant 62371411, InnoHK initiative, the Government of the HKSAR, Laboratory for AI-Powered Financial Technologies.

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

# Appendix

## A   Efficiency of Different Models

### A.1   Token Consumption

| Model | Method | Dataset | | | | | |
|-------|--------|---------|------|-----------|------|-------|-------|
| | | Date | CSQA | StrategyQA | AuQA | SVAMP | GSM8K |
| GPT-3.5 | CoT | 174K/19K | 959K/77K | 476K/67K | 178K/45K | 945K/76K | 1.3M/169K |
| | Complex | 169K/20K | 1.4M/81K | 833K/103K | 523K/46K | 2.5M/96K | 3.6M/195K |
| | ADPSC | 727K/86K | 6.1M/351K | 3.6M/490K | 2.7M/247K | 8.8M/261K | 14.3M/716K |
| | SC | 1.7M/194K | 14.4M/8.3M | 8.3M/1.1M | 5.2M/452K | 25.5M/703K | 36.3M/1.6M |
| | RCI | 501K/64K | 4.5M/263K | 2.4M/214K | 1.4M/122K | 6.6M/211K | 10.2M/469K |
| | LeCo+CoT | 386K/35K | **2.0M/125K** | **1.1M/127K** | 406K/81K | **1.9M/136K** | **2.5M/337K** |
| | LeCo+Complex | **363K/35K** | 3.0M/151K | 1.9M/182K | 1.2M/104K | 5.1M/170K | 8.2M/394K |
| GPT-4 | CoT | 174K/19K | 959K/76K | 476K/58K | 178K/33K | 945K/72K | 1.3M/163K |
| | Complex | 169K/20K | 1.4M/77K | 833K/94K | 523K/40K | 2.5M/92K | 3.6M/177K |
| | ADPSC | 721K/92K | 6.2M/350K | 3.7M/466K | 3.0M/244K | 10.8M/318K | 14.1M/684K |
| | SC | 1.7M/209K | 14.4M/791K | 8.3M/1.0M | 5.2M/405K | 25.5M/701K | 36.3M/1.4M |
| | RCI | 393K/42K | 3.5M/186K | 2.3M/226K | 1.7M/134K | 9.1M/261K | 9.8M/475K |
| | LeCo+CoT | 357K/30K | **2.0M/110K** | **999K/99K** | **388K/58K** | **1.9M/126K** | **2.5M/326K** |
| | LeCo+Complex | **341K/34K** | 3.0M/149K | 1.8M/167K | 1.2M/85K | 5.5M/168K | 7.4M/334K |

Table 8: Average consumed in/out tokens with OpenAI models.

| Model | Method | Dataset | | | | | | |
|-------|--------|---------|-------|----------|------|------|------------|------------|
| | | Algebra | Count | Geometry | Iter | Num | Prealgebra | Precaculus |
| GPT-3.5 | Complex | 2.9M/254K | 1.2M/96K | 1.2M/113K | 2.2M/295K | 1.3M/117K | 2.1M/146213 | 1.3M/165K |
| | RCI | 8.5M/701K | 3.7M/305K | 4.1M/321K | 7.7M/658K | **4.1M/392K** | 6.9M/426K | 4.4M/491K |
| | ADPSC | 15.5M/1.5M | 6.2M/608K | 6.7M/744K | 15.0M/1.9M | 7.7M/721K | 14.7M/1.1M | 11.6M/1.5M |
| | SC | 28.9M/2.6M | 11.6M/934K | 12.0M/10.8M | 22.2M/2.7M | 13.1M/1.2M | 21.3M/1.5M | 13.5M/1.9M |
| | LeCo+Complex | **7.4M/627K** | **3.3M/273K** | **3.4M/309K** | **6.9M/860K** | 4.2M/349K | **5.5M/361K** | **4.1M/483K** |
| GPT-4 | Complex | 2.9M/216K | 1.2M/86K | 1.2M/96K | 2.2M/241K | 13.1M/104K | 2.1M/124K | 1.3M/144K |
| | RCI | 10.4M/613K | 4.3M/267K | 4.6M/283K | 8.5M/626K | 4.9M/323K | 7.4M/325K | 5.0M/446K |
| | ADPSC | 16.7M/1.4M | 8.4M/692K | 8.3M/719K | 19.3M/2.1M | 10.1M/880K | 12.0M/786K | 11.4M/1.3M |
| | SC | 29.0M/1.9M | 11.6M/895K | 12.0M/1.1M | 22.2M/2.3M | 13.1M/1.1M | 21.4M/1.3M | 13.5M/1.5M |
| | LeCo+Complex | **7.4M/515K** | **3.2M/227K** | **3.5M/270K** | **7.2M/720K** | **3.6M/273K** | **5.0M/274K** | **4.2M/432K** |

Table 9: Average consumed in/out tokens on MATH dataset with OpenAI models.

| Models | Methods | GSM8K | Math | | | | | | |
|--------|---------|-------|---------|-------|----------|------|-----|------------|------------|
| | | | Algebra | Count | Geometry | Iter | Num | Prealgebra | Precaculus |
| DeepSeek | Complex | 3.8M/275K | 2.8M/376K | 1.1M/144K | 1.1M/159K | 2.1M/425K | 1.2M/189K | 2.0M/195K | 1.3M/272K |
| | LeCo+Complex | 8.7M/589K | 6.2M/878K | 2.7M/353K | 2.8M/410K | 5.4M/1.1M | 3.1M/458K | 4.6M/457k | 3.4M/708K |

Table 10: Average consumed in/out tokens on MATH and GSM8K datasets with DeepSeek model.

### A.2   Average Iterations Numbers by Different Methods and Models

Table 11 and 12 present the average iteration numbers on arithmetic reasoning, common-sense reasoning, logical reasoning, and complex mathematical reasoning using OpenAI models. Table 13 illustrates the average iteration numbers on the GSM8K and MATH datasets using the DeepSeek model.

| Model | Method | Dataset | | | | | | Avg. |
|---|---|---|---|---|---|---|---|---|
| | | Date | CSQA | StrategyQA | AuQA | SVAMP | GSM8K | |
| GPT-3.5 | ADPSC | 4.31 | 4.21 | 4.43 | 5.13 | 4.27 | 4.42 | 4.46 |
| | RCI | 2.39 | 2.90 | 2.57 | 3.67 | 2.56 | 2.35 | 2.74 |
| | LECO+CoT | 2.16 | 2.08 | 2.18 | 2.16 | 2.14 | 2.20 | **2.15** |
| | LECO+Complex | 2.11 | 2.08 | 2.17 | 2.43 | 2.24 | 2.29 | 2.22 |
| GPT-4 | ADPSC | 4.28 | 4.32 | 4.56 | 5.44 | 4.39 | 4.21 | 4.53 |
| | RCI | 2.08 | 2.31 | 2.47 | 2.9 | 3.21 | 2.25 | 2.54 |
| | LECO+CoT | 2.00 | 2.02 | 2.05 | 2.08 | 2.05 | 2.05 | **2.04** |
| | LECO+Complex | 2.01 | 2.05 | 2.08 | 2.24 | 2.13 | 2.08 | 2.10 |

Table 11: Average iterations on diverse datasets with OpenAI models.

| Model | Method | Dataset | | | | | | | Avg. |
|---|---|---|---|---|---|---|---|---|---|
| | | Algebra | Count | Geometry | Iter | Num | Prealgebra | Precaculus | |
| GPT-3.5 | ADPSC | 5.36 | 5.92 | 6.21 | 5.84 | 6.76 | 5.59 | 6.36 | 6.01 |
| | RCI | 2.59 | 2.83 | 3.00 | 2.75 | 2.97 | 2.58 | 2.78 | 2.79 |
| | LECO+Complex | 2.52 | 2.83 | 2.81 | 2.91 | 2.78 | 2.42 | 2.94 | **2.74** |
| GPT-4 | ADPSC | 6.44 | 7.22 | 5.91 | 7.70 | 8.63 | 5.03 | 8.38 | 7.04 |
| | RCI | 3.31 | 3.41 | 3.51 | 3.41 | 3.43 | 3.27 | 3.29 | 3.38 |
| | LECO+Complex | 2.47 | 2.75 | 2.9 | 2.79 | 2.63 | 2.31 | 2.81 | **2.66** |

Table 12: Average iterations on MATH dataset with OpenAI models.

| Models | Methods | GSM8K | MATH | | | | | | | Avg. |
|---|---|---|---|---|---|---|---|---|---|---|
| | | | Algebra | Count | Geometry | Iter | Num | Prealgebra | Precaculus | |
| DeepSeek | LECO+Complex | 2.25 | 2.22 | 2.44 | 2.46 | 2.52 | 2.45 | 2.25 | 2.59 | 2.40 |

Table 13: Average iterations on MATH and GSM8K datasets with DeepSeek model.

| Models | Methods | Dataset | |
|---|---|---|---|
| | | GSM8K | StrategyQA |
| gpt-3.5-turbo-0613 | Early Stop | 8.0M/367.6K | 1.7M/132.7K |
| | LeCo | 8.2M/393.8K | 1.9M/181.9K |
| gpt-4 | Early Stop | 7.0M/315.7K | 1.7M/162.3K |
| | LeCo | 7.4M/334.2K | 1.8M/167.3K |

Table 14: Average Token Consumption on GSM8K and StrategyQA of Early-stop LECO

# B  Details of Early Stop LECO

## B.1  Algorithm of Early stop LECO

As presented in Algorithm 20, firstly, we sample the entire dataset according to a certain proportion, obtaining distributions of correct and incorrect solutions. Leveraging the normal distribution traits of incorrect responses, we utilize the positive 1-$\sigma$ value as the threshold. For the remaining data, if its solution score surpasses the threshold, we accept this answer outright; otherwise, we resort to the standard LECO method for reconsideration.

## B.2  Token Consumption and Iteration Number of Early Stop LECO

Table 14 and 15 presents the average token consumptions and average iteration numbers on the GSM8K and StrategyQA datasets using OpenAI models via early-stop LECO.

---

**Algorithm 2** Early Stop of LECO

---

**Require:** input questions $x$, model $M$, demonstration $Demo_x$, standard $\text{LECO}(*)$, sample amount $R$, solution score $sln\_score(*)$, normalize function $norm(*)$
1: sample_correct_set $C = \varnothing$, sample_incorrect_set $E = \varnothing$     $\triangleright$ Initialize sample score set
2: **for** $x_s \in 0, \ldots, R$ **do**                                         $\triangleright$ Sample Stage
3:      $y_{t_s} = \text{LECO}(x_s, M, Demo_x)$         $\triangleright$ The subscript $_s$ represents the sampling stage
4:      **if** $y_{t_s}$ is correct **then**
5:          $C \leftarrow C \cup sln\_score(y_{t_s})$
6:      **else**
7:          $E \leftarrow E \cup sln\_score(y_{t_s})$
8:      **end if**
9: **end for**
10: $\mu\_incorrect, \sigma\_incorrect = norm(E)$
11: threshold $t = \mu\_incorrect + \sigma\_incorrect$
12: **for** $x_{ns} \in R + 1, \ldots$ **do**                              $\triangleright$ Early Stop Stage
13:      $y_{0_{ns}} = \mathcal{M}(x_{ns}, Demo_x)$        $\triangleright$ The subscript $_{ns}$ represents the remaining part.
14:      **if** $sln\_score(y_{0_{ns}}) > t$ **then**
15:          $y_{t_{ns}} = y_{0_{ns}}$
16:      **else**
17:          $y_{t_{ns}} = \text{LECO}(x_{0_{ns}}, M, Demo_x, y_{0_{ns}})$
18:      **end if**
19: **end for**
20: **return** $y_t$

---

| Models | Methods | Dataset | |
|---|---|---|---|
| | | GSM8K | StrategyQA |
| gpt-3.5-turbo-0613 | Early Stop | 2.16 | 2.11 |
| | LeCo | 2.39 | 2.17 |
| gpt-4 | Early Stop | 2.03 | 2.06 |
| | LeCo | 2.08 | 2.08 |

Table 15: Average Iterations on GSM8K and StrategyQA of Early-stop LECO

| K | 1 | 3 | 5 |
|---|---|---|---|
| Complex | 81.8 | 80.89 | 83 |
| LeCo + Complex | 82.83 | 82.33 | 83.87 |
| | (+1.03) | (+1.44) | (+0.87) |

Table 16: Settings of Hyperparameter $K$

| $\tau$ | 0.1 | 0.2 | 0.3 | 0.4 | 0.5 |
|---|---|---|---|---|---|
| Complex | 81.16 | 80.98 | 80.89 | 82.46 | 83.03 |
| LeCo+Complex | 82.46 | 82.24 | 82.33 | 83.88 | 83.84 |
| | (+1.3) | (+1.26) | (+1.44) | (+1.42) | (+0.81) |

Table 17: Settings of Hyperparameter $\tau$

## C   Hyperparameter Settings

We compared the experimental results under different settings and found that our method is relatively insensitive to hyperparameters, such as $K$ and $\tau$. We attach the experimental results of GPT-3.5 on GSM8K as follows.

Table 16 and Table 17 present the settings of hyperparameter $K$ and $\tau$.

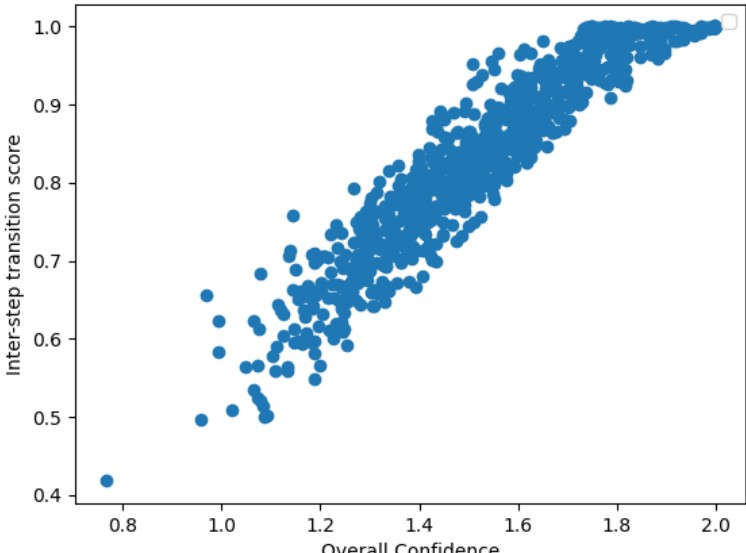

Figure 4: The relation between overall confidence and inter-step transition scores

In the design of the transition score, the parameter $K$ determines the usage of several initial tokens, hence the value of $K$ can not be very large and we set $K$ varying from 1 to 5.

In the design of the divergence score, the parameter $\tau$ is used to rescale the KL divergence to a reasonable range and helps the divergence score to show significant performance. When $\tau$ exceeds 0.5 in the logarithmic function, the divergence diminishes to negligible values, such as 0.002 or 0.004, which fail to capture the desired differences. Consequently, our study focuses on the impact of $\tau$ within the range of 0.1 to 0.5.

The results, as depicted in the tables, reveal a consistent improvement, indicating the robustness of our method to these parameter.

## D  Preliminary Experiments

We draw the scatter plot of the relationship between the overall confidence score and inter-step transition score for 1000 reasoning steps. As shown in Fig4, it's obvious that the overall confidence and inter-step transition scores are highly positively correlated.

## E  Case Study of LECO

Table 18, 19 and 20 lists the specific cases of reasoning results via different methods on GSM8K, StrategyQA, and MATH datasets.

| | |
|---|---|
| Question | A shoe store was having a weekend sale on a brand of popular tennis shoes. On Friday the store sold 14 pairs of tennis shoes. The next day they sold double that number of shoes. On the last day of the sale they sold one-half the amount that they did the day before, but six people returned their pairs because they didn't fit. How many tennis shoes were sold by the end of the sale? |
| Gold Answer | 50.0 |
| CoT | Let's think step by step
Step 1: On Friday, the store sold 14 pairs of tennis shoes.
Step 2: The next day, they sold double that number, which is 14 x 2 = 28 pairs of tennis shoes.
Step 3: On the last day, they sold one-half the amount of the day before, which is 28 / 2 = 14 pairs of tennis shoes.
Step 4: But 6 pairs were returned, so the total number of tennis shoes sold is 14 + 14 - 6 = 22 pairs.
Step 5: The answer is \boxed{22}. |
| Complex | Let's think step by step
Step 1: On Friday, the store sold 14 pairs of tennis shoes.
Step 2: The next day, they sold double that number, so they sold 14 x 2 = 28 pairs of tennis shoes.
Step 3: On the last day, they sold one-half the amount they did the day before, so they sold 28 / 2 = 14 pairs of tennis shoes.
Step 4: However, six people returned their pairs, so the total number of pairs sold is 14 + 14 - 6 = 22 pairs of tennis shoes.
Step 5: Since each pair of tennis shoes represents one unit, the store sold 22 tennis shoes by the end of the sale.
Step 6: The answer is \boxed{22}. |
| RCI | The previous answer is incorrect because it only calculates the number of pairs of tennis shoes sold, not the total number of tennis shoes. To find the total number of tennis shoes, we need to multiply the number of pairs by 2.
Let's correct the answer:
Step 1: On Friday, the store sold 14 pairs of tennis shoes.
Step 2: The next day, they sold double that number, so they sold 14 x 2 = 28 pairs of tennis shoes.
Step 3: On the last day, they sold one-half the amount they did the day before, so they sold 28 / 2 = 14 pairs of tennis shoes.
Step 4: However, six people returned their pairs, so the total number of pairs sold is 14 + 14 - 6 = 22 pairs of tennis shoes.
Step 5: Since each pair of tennis shoes represents two units (one for the left foot and one for the right foot), the store sold 22 x 2 = 44 tennis shoes by the end of the sale.
The answer is \boxed{44}. |
| ADPSC | Let's think step by step
Step 1: On Friday, the store sold 14 pairs of tennis shoes.
Step 2: On the next day, they sold double that number, so they sold 14 * 2 = 28 pairs of tennis shoes.
Step 3: On the last day, they sold half the amount they did the day before, so they sold 28 / 2 = 14 pairs of tennis shoes.
Step 4: However, six people returned their pairs, so the total number of pairs sold is 14 + 14 - 6 = 22 pairs of tennis shoes.
Step 5: Since each pair consists of two shoes, the total number of tennis shoes sold is 22 * 2 = 44 tennis shoes.
Step 6: The answer is 44. |
| SC | Let's think step by step
Step 1: On Friday, the store sold 14 pairs of tennis shoes.
Step 2: The next day, they sold double that number, so they sold 14 x 2 = 28 pairs of tennis shoes.
Step 3: On the last day, they sold one-half the amount as the day before, so they sold 28 / 2 = 14 pairs of tennis shoes.
Step 4: However, 6 people returned their pairs, so the total number of pairs sold is 14 + 14 - 6 = 22 pairs of tennis shoes.
Step 5: Since each pair of tennis shoes consists of 2 shoes, the total number of tennis shoes sold is 22 x 2 = 44 tennis shoes.
Step 6: The answer is 44. |
| COT+LeCo | Let's think step by step
Step 1: On Friday, the store sold 14 pairs of tennis shoes.
Step 2: On Saturday, they sold double that number, so 14 x 2 = 28 pairs of tennis shoes.
Step 3: On the last day, they sold one-half the amount of the day before, so 28 / 2 = 14 pairs of tennis shoes.
Step 4: But 6 pairs were returned, so 14 - 6 = 8 pairs of tennis shoes were sold on the last day.
Step 5: In total, they sold 14 + 28 + 8 = 50 pairs of tennis shoes.
Step 6: The answer is \boxed{50}. |
| Complex+LeCo | Let's think step by step
Step 1: On Friday, the store sold 14 pairs of tennis shoes.
Step 2: The next day, they sold double that number, so they sold 14 x 2 = 28 pairs of tennis shoes.
Ste p3: On the last day, they sold one-half the amount they did the day before, so they sold 28 / 2 = 14 pairs of tennis shoes.
Step 4: However, six people returned their pairs, so the total number of tennis shoes sold is 14 + 28 + 14 - 6 = 50 pairs of tennis shoes.
Step 5: The answer is \boxed{50} pairs of tennis shoes. |

Table 18: Case Study of LeCo on GSM8K by GPT-3.5-Turbo

| Question | Could JPMorgan Chase give every American $10? |
|---|---|
| Gold Answer | Yes |
| CoT | Let's think step by step.
Step 1: JPMorgan Chase is a bank.
Step 2: Banks have a lot of money.
Step 3: There are approximately 330 million Americans.
Step 4: $10 multiplied by 330 million is 3.3 billion.
Step 5: JPMorgan Chase is a very large bank, but it may not have enough money to give $10 to every American.
Step 6: Thus, JPMorgan Chase may not be able to give every American $10. Step 7: So the answer is \boxed{no}. |
| Complex | Let's think step by step. Step 1: JPMorgan Chase is a multinational investment bank and financial services company.
Step 2: JPMorgan Chase has a market capitalization of over $400 billion.
Step 3: The population of the United States is over 330 million.
Step 4: To give every American $10, JPMorgan Chase would need to distribute over $3.3 billion.
Step 5: JPMorgan Chase's market capitalization is significantly smaller than the amount needed to give every American $10.
Step 6: Thus, JPMorgan Chase could not give every American $10.
Step 7: So the answer is \boxed{no} |
| RCI | I apologize for the mistake in my previous answer.
Upon reviewing the information, it is clear that JPMorgan Chase would not be able to give every American $10.
The amount required to give $10 to each of the approximately 331 million Americans would be $3.31 billion,
which is a significant amount even for a large financial institution like JPMorgan Chase.
Therefore, the correct answer is \boxed{no}. |
| ADPSC | Let's think step by step.
Step 1: JPMorgan Chase is a multinational investment bank and financial services company.
Step 2: The current population of the United States is approximately 331 million people.
Step 3: If JPMorgan Chase were to give every American $10, they would need to distribute a total of $3.31 billion.
Step 4: JPMorgan Chase's total assets as of 2020 were $3.42 trillion.
Step 5: $3.31 billion is a very small fraction of JPMorgan Chase's total assets.
Step 6: Thus, JPMorgan Chase could give every American $10.
Step 7: So the answer is \boxed{yes}. |
| SC | Let's think step by step.
Step 1: JPMorgan Chase is a multinational investment bank and financial services company.
Step 2: The current population of the United States is approximately 331 million people.
Step 3: If JPMorgan Chase were to give every American $10, they would need to distribute a total of $3.31 billion.
Step 4: JPMorgan Chase's total assets as of 2020 were $3.42 trillion.
Step 5: $3.31 billion is a very small fraction of JPMorgan Chase's total assets.
Step 6: Thus, JPMorgan Chase could give every American $10.
Step 7: So the answer is \boxed{yes} |
| COT+LeCo | Let's think step by step
Step 1: JPMorgan Chase is a bank.
Step 2: Banks have a lot of money.
Step 3: Thus, JPMorgan Chase could potentially give every American $10.
Step 4: So the answer is \boxed{yes} |
| Complex+LeCo | Let's think step by step
Step 1: JPMorgan Chase is a multinational investment bank and financial services company.
Step 2: JPMorgan Chase is one of the largest banks in the United States.
Step 3: The population of the United States is over 300 million people.
Step 4: To give every American $10, JPMorgan Chase would need to distribute over $3 billion.
Step 5: JPMorgan Chase's total assets are over $3 trillion.
Step 6: Thus, JPMorgan Chase could give every American $10.
Step 7: So the answer is \boxed{yes} |

Table 19: Case Study of LeCo on StrategyQA by GPT-3.5-Turbo

| | |
|---|---|
| Question | Five balls are numbered with the integers 1 through 5 and placed in a jar.
Three are drawn without replacement.
What is the probability that the sum of the three integers on the balls is odd?
Express your answer as a common fraction. |
| Gold Answer | $\frac{2}{5}$ |
| Complex | Let's think step by step
Step 1: There are two cases to consider: either all three integers are odd, or exactly two of the integers are odd.
Step 2: Case 1: If all three integers are odd, there are $\binom{3}{3} = 1$ ways to choose the odd integers.
Step 3: There are $\binom{5}{3} = 10$ total ways to choose three integers from the five.
Step 4: Case 2: If exactly two of the integers are odd, there are $\binom{3}{2} = 3$ ways to choose the odd integers and $\binom{2}{1} = 2$ ways to choose the even integer.
Step 5: There are $\binom{5}{3} = 10$ total ways to choose three integers from the five.
Step 6: So, the probability that the sum of the three integers is odd is $\frac{1+3\cdot2}{10} = \frac{1+6}{10} = boxed\{\frac{7}{10}\}$.
Step 7: The answer is \boxed{$\frac{7}{10}$}. |
| RCI | A: Let's think step by step
Step 1: There are a total of $\binom{5}{3} = 10$ ways to choose 3 balls out of 5.
Step 2: We need to count the number of ways to choose 3 balls such that the sum of their integers is odd.
Step 3: The sum of three integers will be odd if and only if there are an odd number of odd integers among the three chosen balls.
Step 4: There are $\binom{3}{1} = 3$ ways to choose 1 odd integer and 2 even integers.
Step 5: There are $\binom{2}{3} = 0$ ways to choose 3 odd integers.
Step 6: There are $\binom{3}{2} = 3$ ways to choose 2 odd integers and 1 even integer.
Step 7: So, the total number of ways to choose 3 balls such that the sum of their integers is odd is $3 + 0 + 3 = 6$.
Step 8: Therefore, the probability is $\frac{6}{10} = boxed\{\frac{3}{5}\}$.
Step 9: The answer is \boxed{$\frac{3}{5}$}. |
| ADPSC | Let's think step by step
Step 1: There are two cases to consider: either all three integers are odd, or two are odd and one is even.
Step 2: Case 1: If all three integers are odd, there are $\binom{3}{3} = 1$ ways to choose the odd integers.
Step 3: There are 5 odd integers to choose from, so the probability in this case is $\frac{1}{\binom{5}{3}} = \frac{1}{10}$.
Step 4: Case 2: If two integers are odd and one is even, there are $\binom{3}{2} \cdot \binom{2}{1} = 3 \cdot 2 = 6$ ways to choose the integers.
Step 5: There are 3 odd integers and 2 even integers to choose from, so the probability in this case is $\frac{6}{\binom{5}{3}} = \frac{6}{10} = \frac{3}{5}$.
Step 6: The total probability is $\frac{1}{10} + \frac{3}{5} = \frac{1}{10} + \frac{6}{10} = \frac{7}{10}$.
Step 7: The answer is \boxed{$\frac{7}{10}$} |
| SC | Let's think step by step
Step 1: There are two cases to consider: either all three integers are odd, or two are odd and one is even.
Step 2: Case 1: If all three integers are odd, there are $\binom{3}{3} = 1$ ways to choose the odd integers.
Step 3: There are 5 odd integers to choose from, so the probability in this case is $\frac{1}{\binom{5}{3}} = \frac{1}{10}$.
Step 4: Case 2: If two integers are odd and one is even, there are $\binom{3}{2} \cdot \binom{2}{1} = 3 \cdot 2 = 6$ ways to choose the integers.
Step 5: There are 3 odd integers and 2 even integers to choose from, so the probability in this case is $\frac{6}{\binom{5}{3}} = \frac{6}{10} = \frac{3}{5}$.
Step 6: The total probability is $\frac{1}{10} + \frac{3}{5} = \frac{1}{10} + \frac{6}{10} = \frac{7}{10}$.
Step 7: The answer is \boxed{$\frac{7}{10}$} |
| Complex+LeCo | Let's think step by step
Step 1: There are two cases to consider: the sum is odd if either all three numbers are odd or two numbers are odd and one number is even.
Step 2: Case 1: All three numbers are odd. There are 3 odd numbers (1, 3, 5) and 2 even numbers (2, 4) to choose from.
Step 3: The probability of choosing 3 odd numbers is $\frac{3}{5} \cdot \frac{2}{4} \cdot \frac{1}{3} = \frac{1}{10}$.
Step 4: Case 2: Two numbers are odd and one number is even. There are 3 odd numbers and 2 even numbers to choose from.
Step 5: The probability of choosing 2 odd numbers and 1 even number is $\frac{3}{5} \cdot \frac{2}{4} \cdot \frac{2}{3} + \frac{3}{5} \cdot \frac{2}{4} \cdot \frac{2}{3} + \frac{3}{5} \cdot \frac{2}{4} \cdot \frac{2}{3} = \frac{3}{10}$.
Step 6: The total probability is $\frac{1}{10} + \frac{3}{10} = boxed\{\frac{2}{5}\}$.
Step 7: The answer is \boxed{$\frac{2}{5}$}. |

Table 20: Case Study of LECO on the MATH dataset using GPT-3.5-Turbo.

