# OpenReview forum: "Learning From Correctness Without Prompting Makes LLM Efficient Reasoner"
_colmweb.org/COLM/2024/Conference — COLM_

### Official Review · Reviewer_cvVx · 2024-05-08

**Rating:** 5
**Confidence:** 4
**Ethics Flag:** 1

**Summary:**

This paper introduces a reasoning paradigm called Learning from Correctness (LECO), aiming to enhance the performance of Large Language Models (LLMs) in reasoning tasks. The LECO framework employs a multi-step reasoning paradigm, first computing confidence scores for each reasoning step, which include considerations such as average token scores, step divergence scores, and step transition scores. Subsequently, LECO identifies the step with the lowest confidence as a potential error step and treats all preceding steps as "correct." Then, LECO incorporates these "correct" steps as part of the input, refining the search space through iterative reasoning processes until obtaining the final answer or reaching a stopping condition. Experimental results demonstrate that this approach not only reduces reliance on external interventions but also enhances the model's performance in multi-step reasoning tasks while reducing token consumption.

**Questions To Authors:**

1. In Algorithm 1, when the iteration variable t ranges from 0 to t, what does x_(t-1) represent when t=0?
2. In step 7 of Algorithm 1, what does the procedure mean? Does it involve concatenating the two strings? Further elaboration on this seems needed in this paper.
3. Figure 3 appears to lack clarity.
4. It seems that the step divergence score is not effectively functioning across the three confidence scores as indicated in Table 5. Could you provide a more in-depth analysis of this observation? Moreover, could you elucidate why the distribution is expected to approximate a uniform distribution? Additionally, while I understand that the step divergence score may to some extent reflect the distribution of token probabilities, how does it specifically capture higher token probabilities as opposed to lower ones? (the higher the token probabilities, the bigger the step divergence score?)
5. In the discussion of the inter-step transition score, you mentioned two key insights from your pre-experimentation phase. However, I apologize for not noticing where your preliminary experiment was referenced. It's possible that I missed it, but could you please indicate its location in the paper?
6. Which models are the green and red markers in the Table compared against? Please mention this in the caption.
7. I observed that the improvement on DeepseekMath-7B is not as significant compared to the enhancement seen in the GPT series. Does this suggest that your approach requires a higher standard of model performance itself? Given that your methodology entails the model's capability to generate the correct steps as assumed, would it not imply that, with lower model performance, the selection process might involve choosing steps that are less incorrect rather than truly correct?

**Reasons To Accept:**

This article differs from previous approaches like learning from errors, as it adopts a method of identifying correct steps in the reasoning process and progressively obtaining the correct answer through these steps. Importantly, it does not require training and is applicable to black-box models that are not publicly disclosed.

**Reasons To Reject:**

1.	https://arxiv.org/pdf/2404.14963 This research has significantly improved the performance of GPT-4 in GSM8K, achieving up to a 97% enhancement compared to prior work.
2.	The experimental design in the paper is not sufficiently comprehensive. For instance, it lacks adequate baseline tests and comparisons with state-of-the-art methods. Additionally, the statistical significance of the experimental results is insufficient.
3.	Is it reasonable to assume that using the level of confidence as a judgment of the correctness of the reasoning process is valid? If a model generates illusions, errors, or unfaithfulness, does its confidence necessarily remain low? Please provide some relevant theoretical explanations.

---

> ### Author Rebuttal · Authors · 2024-05-31
>
> We sincerely appreciate the reviewer and address your concerns as follows:
>
> ---
>
> **R4A1:**  Comparison with other works
>
> We would like to clarify that the paper mentioned by the reviewer (DUP) was posted to arxiv after the CoLM submission date, and API versions of GPT may differ. Additionally, there are also key differences between DUP and LeCo. 1) DUP needs elaborate prompts while LeCo eliminates the need for external information; 2) DUP is hard to be used with open-source models.
> Moreover, as stated in R3A2, our goal is not to achieve new SOTA performance.
>
> **R4A2:** Confidence
>
> Previous works [1,2] also use confidence as the judgment on reasoning tasks. Our experimental results show that 65% of incorrect reasoning steps are the steps with the lowest confidence.
>
> **R4A3:** Step divergence
>
> 1. The step divergence score should be evaluated within the overall design of LeCo, as its standalone effect, focusing solely on within-step distribution, is less impactful.
> 2. We tested various distributions( Laplace, Rayleigh distribution...), and the normal distribution best matched the data patterns, leading to its selection.
> 3. Step divergence evaluates the uniformity within a step rather than individual tokens.  The divergence score for step [.85, .83, .86, .84] is lower than that for [.95, .63, .86, .94]
>
> **R4A4:** Preliminary findings
>
> 1. We draw the scatter plot of the relationship between the overall confidence score and transition confidence score, and find they are positively correlated. We will attach the figure to the appendix.
> 2. It is concluded through observations.
>
> **R4A5:** Model capacity
>
> 1. LeCo prefers stronger LLMs while it also can be used on less-powerful models.
> 2. 100 sampled solutions reveal that LeCo identified the first error step 62% of the time with DeepSeek and 65% with GPT-3.5. DeepSeek's slightly lower accuracy and consistent replies, often identical for 'demonstration' and 'demonstration + several correct steps' inputs, may explain its less significant improvement.
>
> **R4A6:** Typo and presentation
>
> Thanks for pointing out the typo and presentation problems. We will revise them accordingly in the revised version.
>
> ---
>
> References:
>
> [1] Generating with Confidence: Uncertainty Quantification for Black-box Large Language Models ArXiv 2023
>
> [2] Confidence Matters: Revisiting Intrinsic Self-Correction Capabilities of Large Language Models. ArXiv 2024
>
> [3] Self-Refine: Iterative Refinement with Self-Feedback. NeurIPS 2023

---

> > ### Author Response · Authors · 2024-06-05
> > **Follow-up Response**
> >
> > We would be delighted to discuss any further questions or clarifications regarding our work. The following are some supplementary details.
> >
> > ---
> >
> > **Comprehensive Experiments:**
> >
> >
> > We think our experiments are sufficient based on following facts: 1) we evaluated LeCo with different demonstrations (i.e. CoT and Complex), both of them confirming the effectiveness of LeCo; 2) we compared to popular baselines (i.e. SC, ADPSC) and learning-from-error methods (e.g. RCI). Your suggestion work, namely DUP, is published after the submission deadline of CoLM. Besides, we actually attempted other earlier works achieving predominant reasoning performance, such as PHP[4]  and CSV[5], but we did not report the scores. This is because we find these prompting-based methods are extremely sensitive. Adopting the same prompts in their works with the GPT API version we used, we cannot consistently reproduce their reported performance across several different runs.
> >
> > On the other hand, we do not think LeCo is the same kind of work against the prompting-based methods. Although we both aim at improving the multi-step reasoning abilities, our method is a more labor-efficient, token-efficient, and transferable one. We did not need to dive ourselves into designing prompts when encountering a new task or using a new API/model.
> >
> > We admit that LeCo might underperform than the methods using elaborated prompts in some scenarios. We still would like to highlight the superiority of LeCo on transferability and cost-saving.
> >
> > **Confidence:**
> >
> > Following  [1], our method also employs confidence measures based on logits, which have demonstrated effectiveness in enhancing generation quality.
> >
> > We consider a step is incorrect according to the following perspectives: 1) Calculation errors, 2) Exaggeration of reasoning conditions, and 3) Fabrication of non-existent information. Our experimental results indicate that 65% of incorrect reasoning steps correspond to those with the lowest confidence levels, demonstrating the effectiveness of our algorithm.
> >
> > **Model Capacity:**
> >
> > It is widely recognized that the capabilities of Large Language Models (LLMs) play a crucial role in their performance across various tasks. For example, studies [2,6,7] use LLM evaluation to refine original solutions, making it challenging to conduct such evaluations on open-source models.
> >
> > **Step 7 in the algorithm:**
> >
> > LeCo first finds the earliest error step, then considers all previous steps as correctness and splices them into the last reply. Figure 1 shows this process; For instance, if an error occurs in step 2, the next-round reasoning will start at step 2 with the identified correct steps unchanged.
> >
> > **Preliminary experiments:**
> >
> > To demonstrate “2) These initial heading tokens were also the most likely to change across different program runs.”, we present 2  partial solutions of the same question with different seeds.
> >
> > > Q: Out of the 200 Grade 5 students, 2/5 are boys and 2/3 of the girls are in the girl scout. How many girls are not in the girl scout?
> >
> > > A1:  Step 2: The number of boys is 2/5 * 200 = 80. Step 3: The number of girls is 200 - 80 = 120. Step 4: Now, let's find the number of girls in the girl scout.
> >
> > > A2:  Step 2: If 2/5 of the students are boys, then there are 2/5 * 200 = 80 boys. Step 3: The remaining students are girls, so there are 200 - 80 = 120 girls. Step 4: If 2/3 of the girls are in the girl scout, then there are 2/3 * 120 = 80 girls in the girl scout.
> >
> > It’s easy to notice that these initial heading tokens were also the most likely to change across different program runs.
> >
> > **About Q1,Q3 and Q6:**
> >
> > For Q1, thanks for pointing out the typo, we will revise the starting index.
> >
> > For Q3, we would like to clarify that the figure represents the score distribution. The sample set used in early stop LeCo is drawn from the test set to help determine the threshold. The remaining portion of the test set is then assessed based on this threshold, as explained in the Further Analysis section. The figure is plotted to demonstrate that the sample set is appropriate for determining the threshold since both the sample set and the test set exhibit similar distributions. We will add more detailed legends and annotations to Figure 3 for better clarity
> >
> > For Q6, The markers represent comparisons with base CoT methods and Complex-CoT methods correspondingly.
> >
> > ---
> >
> > We sincerely thank the reviewer again for your valuable suggestions. If you have any further questions, please let us know and we will be happy to discuss them with you further.
> >
> > References:
> >
> > [4] Progressive-Hint Prompting Improves Reasoning in Large Language Models Arxiv 2023
> >
> > [5] Solving challenging math word problems using gpt-4 code interpreter with code-based self-verification ICLR 2024
> >
> > [6] Self-Refine: Iterative Refinement with Self-Feedback.  NeurIPS 2024
> >
> > [7] Teaching Large Language Models to Self-Debug. ICLR 2024

---

> ### Author Response · Authors · 2024-06-07
> **Further discussions are appreciated!**
>
> Dear Reviewer cvVx,
>
> We sincerely appreciate your time to review our submission and provide valuable comments. We have carefully considered all of your concerns and tried to resolve them in our rebuttal and last response. Your constructive feedback will greatly help use improve the quality of the work.
>
> As the deadline of discussion period is apporaching, we notice your rating remains at 4, leaning torwards the rejection. We would really appreciate it if you have read our response and let us konw whether the previous responses have addressed your concerns accordingly. If your concerns have not been well resolved, could you please let us know our remaining concerns so that we have the opportunity to respond before the deadline? We are happy to have any follow-up discussions. If you are satisfied with our response and it truly addresses your concerns, we would really appeciate it if you could consider to increase the rating score.
>
> We understand you are very busy and we really appreciate your time. Looking forward to your further comments and discussions.
>
> Best wishes,
>
> Authors

---

### Official Review · Reviewer_Me19 · 2024-05-09

**Rating:** 6
**Confidence:** 3
**Ethics Flag:** 1

**Summary:**

This paper aims to address some limitations in LLMs, such as hallucination, unfaithful reasoning, and toxic content. In particular, the authors proposed an intrinsic self-correct reasoning framework for LLMs that eliminates the need for human feedback, external tools, and handcraft prompts. The new framework follows a multi-step reasoning paradigm, learning from correctness.

**Questions To Authors:**

----Post Rebuttal----
Some of my previous concerns have been addressed, and thus I would like increase my score.

**Reasons To Accept:**

1. This paper studies an important problem, i.e., improving the reliability and reasoning capability of LLMs. Instead of learning from errors as in existing work, this paper presents a new framework by learning from correctness. The idea is well motivated.

2. Overall, the paper is well organized and clearly written.

3. Extensive results on benchmarks are reported. The proposed LECO framework helps improve the performance in most cases. Ablation studies are provided, and case studies are also presented in the appendix.

**Reasons To Reject:**

1. The confidence score proposed in Eq. (4) is quite heuristic and thus it requires more justifications.

2. Comparing the proposed LECO framework with baselines (such as SC), the improvements are marginal in many cases, according to the results reported in Table 1 and Table 2.

3. The computational costs of LECO and baselines are not discussed and compared in the experiments.

---

> ### Author Rebuttal · Authors · 2024-05-31
>
> We sincerely appreciate your insightful feedback. We will carefully address your concerns in the following points.
>
> ---
>
> **R3A1**  Eq(4):
>
> The *avg_score* reflects the general quality of a step, *trans_score* measures consistency between steps. While the *diver_score* assesses the importance of specific tokens within a step, as detailed in our Methodology. High average and transition scores indicate high-quality steps with continuous logical reasoning. Conversely, a high divergence score suggests non-uniform token distribution within a step, indicating that key tokens, such as those in mathematical calculations, may have low probabilities. Therefore, we aim to keep the divergence score low, assigning it a coefficient of -1 to minimize its impact.
>
> Towards the weights of these scores, we did not conduct the grid-search and simply set them to 1 to avoid introducing too many hyperparameters. Based on this vanilla setting, consistent improvements are also observed in our experiments. Therefore, we believe further benefits can be achieved with an elaborated hyperparameter search.
>
> ---
>
> **R3A2** Improvements:
>
> We acknowledge that the improvements against SC are modest on less complex tasks like commonsense and arithmetic reasoning, but more substantial on challenging datasets such as MATH. This is explained in our manuscript:
> >“ the difficulty of the task correlates positively with the impact of LeCo… The primary reason for this is that the LLM tends to remain their initial reasoning path on the easy problems, offering fewer improvement rooms for LeCo.”
>
> LeCo's primary advantage over SC is its **80%** reduction in token consumption while achieving better performance. Our main contribution lies not in attaining SOTA performance, but in proposing a novel paradigm for leveraging feedback—learning from correctness—to enhance reasoning capabilities while reducing computational costs.
>
> ---
>
> **R3A3** Costs:
>
> Due to the limited space, we replace this comparison in **Appendix A.1**. We also provide a brief discussion about the computation cost in the **last paragraph of “Main Results”**.
>
> |      | CSQA       | GSM8K      |
> | ------- | ------------ | ------------ |
> | SC    | 14.4M/8.3M | 36.3M/1.6M |
> | LeCo  | **3M/151K**   | **8.2M/394K**  |
>
> We can see that LeCo is the **most token-efficient** method among these algorithms, saving about 80% of tokens compared to SC.
>
> ---
>
> We sincerely thank the reviewer again for the valuable suggestions!

---

> > ### Comment · Reviewer_Me19 · 2024-06-04
> >
> > Thanks for the detailed responses. Some of my previous concerns have been addressed, and thus I would like increase my score.

---

> > > ### Author Response · Authors · 2024-06-07
> > > **Thanks for your valuable feedback!**
> > >
> > > Dear Reviewer Me19,
> > >
> > > We sincerely thanks for your valuable feedback and time you dedicated into reviewing our rebuttal. We are glad to know that our response addresses some of your concerns and contributes to increase the score to positive.
> > >
> > > Your suggestions help us a lot to improve the quality of work and we will incorporate the discussion into our revised version.
> > >
> > > The discussion deadline is approaching. If you have any remaining or further concerns, could you please let us know? We are very happy to have further discussions and try to resolve them.
> > >
> > > Really appreciate your time and efforts in reviewing and rebuttal period.
> > >
> > > Best regards,
> > >
> > > Authors.

---

### Official Review · Reviewer_2kVG · 2024-05-11

**Rating:** 8
**Confidence:** 4
**Ethics Flag:** 1

**Summary:**

This paper explores the area of aligning language models but without using human feedback (RLHF), external tools handcrafted prompts (SFT). They show that the resulting model improves reasoning performance. The main idea is learning from correct reasoning steps and based on generation logits and estimation of confidence.

**Questions To Authors:**

- I believe this paper may merit a mention: https://aclanthology.org/2023.findings-emnlp.679.pdf  - the authors of the reviewed paper calculate the confidence via logits  by averaging the token probabilities within a given step + divergence step, while in this paper link I provide they use ppl as a way to correlate with downstream task performance.

- Note this recent paper, very related https://arxiv.org/pdf/2405.00204 (I believe it was posted to arxiv after the CoLM submission date, but I post it here for their awareness.

**Reasons To Accept:**

- The paper is well written and easy to follow.

- Their method (even if simple) and that does not require any external knowledge, helps to improve the reasoning capabilities of the LLMs tested in multiple tasks and benchmarks. The improvements are more prevalent in the MATH dataset.

- I think this paper will be of interest to the research community at the conference.

**Reasons To Reject:**

- The choice of citations is rather odd. For example, if I would have cite a paper on learning from human feedback, I would probably cite https://arxiv.org/abs/2203.02155 and if I have to cite the precursor of language models as we know them today, I would probably cite https://arxiv.org/abs/2005.14165. The authors, however, cite only 2023 papers thus ignoring the actual original works on the topic they refer to.

---

> ### Author Rebuttal · Authors · 2024-05-31
>
> We sincerely appreciate the time and effort you put into reviewing our paper. We appreciate the insightful feedback and address your concerns as follows:
>
> ---
>
> **R2A1** Inappropriate and missing citations
>
> **Response:**  Thanks for pointing out the potential problems regarding our reference. We would like to clarify that reference [1] was mentioned in the first paragraph of the Related Works section. We will include reference [2] and other original works in our revised version.
>
> Reference [3] hypothesizes that a model's familiarity with a prompt's language predicts its effectiveness, demonstrating that lower perplexity prompts yield better performance. The authors also develop a method to expand a small set of manually written prompts through paraphrasing and back-translation, confirming perplexity as a strong predictor of prompt success. In contrast, our work reduces reliance on prompts by calculating confidence using logits.
>
> Reference [4] seeks to enhance the reasoning capabilities of LLMs by exploring different chains of thought and validating individual reasoning steps based on Relevance, Mathematical Accuracy, and Logical Consistency. Constraints are implemented through model verifiers and perplexity checks to ensure high-quality solutions. However, this approach still relies on prompts for verification, similar to our cited works.
>
> In our revised version, we will integrate the discussed research efforts into the corresponding sections.
>
> We sincerely thank the reviewer for their insightful feedback again!
>
> ---
>
> References:
>
> [1] Training language models to follow instructions with human feedback. NeurIPS 2022.
>
> [2] Language Models are Few-Shot Learners. NeurIPS 2020.
>
> [3] Demystifying Prompts in Language Models via Perplexity Estimation. EMNLP 2023, Findings.
>
> [4] General Purpose Verification for Chain of Thought Prompting. ArXiv 2024.

---

> > ### Comment · Reviewer_2kVG · 2024-06-04
> > **Acknowledged on rebuttal by authors**
> >
> > Thanks for adding discussion about the papers I provided. I encourage you consider them in the final version if the paper is accepted.

---

### Official Review · Reviewer_SLMi · 2024-05-11

**Rating:** 7
**Confidence:** 5
**Ethics Flag:** 1

**Summary:**

The paper proposes to improve multi-step reasoning capability of LLM by an intrinsic self-correct framework LECO. Specifically, the proposed self-correct method introduces a step confidence method to check the correctness of each step and append the correct steps to the input so as to enhance the context for multi-step reasoning. To calculate the confidence of each reasoning step, the paper uniformly combines three kind of metrics, that is average token score, step divergence score and inter-step transition score.
The experimental results reported positive improvement by exploiting LECO with COT/Complex on strong LLMs such as GPT-3.5, GPT-4 and DeepSeek. And the ablation study showed the impact of the three metrics on step confidence by correctness. I have suggestion to list the performance of fine-grained ablation study as those of Table 2.
Generally, the paper is well organized and written, easy to follow.
Some suggestions for more experimental results on:
1. the choice of tau in Equation (2)  or why 0.3 is the choice
2. the impact of K in Equation (3) or why 3 is choosen

**Reasons To Accept:**

1. propose a simple and effective self-correct method to enhance the context for multi-step reasoning by introducing step confidence to check the correctness of each step and append the correct steps to the input

**Reasons To Reject:**

1. some experiments supporting the choice of hyper-parameters should be included
(1) the choice of tau in Equation (2)  or why 0.3 is the choice
(2) the impact of K in Equation (3) or why 3 is chosen

---

> ### Author Rebuttal · Authors · 2024-05-31
>
> We sincerely thank the reviewer for the constructive suggestions, which help us to improve the quality of our work, and are pleased that you find our work to be novel and effective.
>
> ---
>
> **R1A1**: The choice of hyperparameters
>
> **Response**: We compared the experimental results under different settings and found that our method is relatively insensitive to hyperparameters, such as *K* and *tau*. We attach the experimental results of GPT-3.5 on GSM8K as follows.
>
> * For *K* :
>
> | K            | 1       | 3       | 5    |
> | -------------- | --------- | --------- | --------- |
> | Complex      | 81.80   | 80.89   | 83.00   |
> | LECO+Complex | 82.83   | 82.33   | 83.97   |
> |             | (+1.03) | (+1.44) | (+0.97) |
>
>
> * For *tau* :
>
> | tau          | 0.1    | 0.2     | 0.3     | 0.4     | 0.5     |
> | -------------- | -------- | --------- | --------- | --------- | --------- |
> | Complex      | 81.16  | 80.98   | 80.89   | 82.86   | 83.03   |
> | LECO+Complex | 82.46  | 82.24   | 82.33   | 83.88   | 83.84   |
> |             | (+1.3) | (+1.26) | (+1.44) | (+1.42) | (+0.81) |
>
>
>
> In the design of the divergence score, the parameter *tau* is used to rescale the KL divergence to a reasonable range and helps the divergence score to show significant performance. When *tau* exceeds 0.5 in the logarithmic function, the divergence diminishes to negligible values, such as 0.002 or 0.004, which fail to capture the desired differences. Consequently, our study focuses on the impact of tau within the range of 0.1 to 0.5. The results, as depicted in the table, reveal a consistent improvement, indicating the robustness of our method to this parameter.
>
> ---
>
> We appreciate the reviewer's insightful feedback and will incorporate this discussion into the revised version of our paper.

---

> > ### Author Response · Authors · 2024-06-07
> > **Further comments and discussions are appreciated!**
> >
> > Dear Reviewer SLMi,
> >
> > We sincerely appreciate your valuable comments and positive feedback.
> >
> > In the previous response, we provided the experimental results on hyperparameters selection to resolve your concerns. The results show that our method is robust to these hyperparameters. We will include this discussion in our revised version.
> >
> > The discussion deadline is approaching. We would really appreciate it if you could let us know if our responses have addressed your concerns satisfactorily. If your concerns have not been resolved, could you please let us know about it so that we have the opportunity to respond before the deadline? We would be happy to have any follow-up discussions and address any additional concerns. We understand you are very busy and we really appreciate your time. We look forward to your valuable feedback.
> >
> > Best wishes,
> >
> > Authors

---

### Decision · Program_Chairs · 2024-07-10

**Decision:**

Accept

**Comment:**

This is an interesting paper that proposes a simple heuristic for progressively improving the accuracy of reasoning step sequences with an LLM.  The technique requires access to the per-token logits, but is otherwise generic, and appears to give useful improvements across datasets and models.  A careful ablation appears to validate the three components of the proposed heuristic score, which otherwise is intuitive but not rigorously justified.

The reviewers gave many helpful suggestions, including relevant references to add.  Moreover, they provide good suggestions to clarify the algorithm exposition, address computational overhead, address hyperparameter sensitivity, and strengthen the justification of the proposed heuristic.  It is expected that the authors will address these in the final version of their paper.